# Statin use and outcome risks according to predicted CVD risk in Korea: A retrospective cohort study

Hae Hyuk Jung ORCID *

Department of Medicine, Kangwon National University Hospital, Kangwon National University School of Medicine, Chuncheon, South Korea

* haehyuk@kangwon.ac.kr

## Abstract

### Background

The validity of cardiovascular disease (CVD) risk calculators in decision for statin therapy has not been fully evaluated at a population level. This study aimed to examine the net benefits of statins according to predicted CVD risk.

### Methods and findings

A cohort of 40 to 79-year-old Korean adults without CVD was generated from the National Health Information Database 2006–2017. Major CVD event rates and all-cause mortality in 58,265 users who initiated statins during 2007–2010 were compared with those in 58,265 non-users matched on propensity scores, from January 1, 2012 through December 31, 2017. Additionally, simulation was performed for the population-based cohort of 659,759 adults. CVD risk was predicted using the 2018 revised Pooled Cohort Equations. In propensity score-matched cohort, the CVD hazard ratios (95% CIs) in occasional, intermittent, and regular statin users were 1.06 (0.93–1.20), 0.82 (0.70–0.97), and 0.57 (0.50–0.64), respectively. The corresponding mortality hazard ratios were 1.01 (0.92–1.10), 0.87 (0.78–0.98), and 0.71 (0.66–0.77), respectively. In stratified analyses, the relative risk reductions were similar, irrespective of age, sex, or predicted CVD risk. Accordingly, absolute risk reductions were greater in higher risk categories. In 6-year follow-up simulation cohorts, regular statin use could reduce 17 CVDs and 28 deaths in 1000 adults with a 10-year risk of $\geq$10.0% vs 10 CVDs and 14 deaths in 1000 with $\geq$2 major risk factors. However, in actual adults with a risk of $\geq$10%, statin use was insufficient and estimated to reduce 3 CVDs and 4 deaths in 1000. Limitations of this study include assessment of medication use based on the prescription data, lack of information on the intensity of statins, and limited generalizability to individuals with very old age or other ethnicity.

### Conclusions

CVD risk calculators were valid in decision-making for primary prevention statin therapy. Proper risk assessment and regular statin use in patients at high predicted risk would reduce outcome risks much more than present in Asian populations.

**Data Availability Statement:** The author cannot legally distribute the data. The data contain potentially identifying or sensitive patient information. The National Health Information Database can be requested from the National

Health Insurance Sharing Service (http://nhiss.nhis.or.kr) and are provided in a "Data analysis room" located within the National Health Insurance Service (https://nhiss.nhis.or.kr/bd/ab/bdaba032eng.do).

**Funding:** The author received no specific funding for this work.

**Competing interests:** The author has declared that no competing interests exist.

## Introduction

Statins are widely used to prevent cardiovascular disease (CVD) events: many studies have shown that statins reduce CVD-related morbidity and mortality in patients with or without prior CVD [1–3]. However, the questions about their harmful effects and beneficiaries have remained unresolved.

In terms of decision-making for statin therapy, it would be important to know whether an absolute risk reduction is worth the potential harms or inconvenience of daily statin use. The relative risk reductions for CVD events in statin vs placebo groups were similar across the majority of primary prevention trials [3–8]. Furthermore, a recent review of statin trials reported that the relative risk reductions were similar across age-, sex-, and other risk-based categories [2]. Given similar relative risk reductions, the absolute benefits of statins will be greater among patients at higher risk. Thus, current guidelines recommend predicting CVD risk with a risk calculator and initiating primary prevention statin therapy in individuals at higher predicted risk [9–11]. However, there remain debates about the risk of statin-related adverse events [12–14], and the validity of CVD risk calculators in decision for statin therapy has not been fully evaluated at a population level.

To assess the net benefit of primary prevention statin therapy according to the predicted risk for CVD, this study examined the associations between statin use status and outcome risks across the predicted risk categories, in a retrospective cohort of Korean adults with no prior CVD.

## Methods

### Participants

This study was conducted from June 2018 to December 2019 using the National Health Information Database (NHID), which is a database for the entire population of Korea managed by the National Health Insurance Service (NHIS) [15]. After the preliminary analysis, the final analysis was performed from February 2019. The Institutional Review Board of Kangwon National University Hospital (IRB File No: KNUH-2018-06-013-001, KNUH-2019-02-001) approved this study and waived the need for informed consent as the data were de-identified prior to analysis. This study is reported as per the Strengthening the Reporting of Observational Studies in Epidemiology (STROBE) for cohort study (S1 Checklist).

A propensity score-matched cohort was generated from the NHID, to compare the outcomes in statin users with those in nonusers. First, one million adults were randomly selected from 7.13 million citizens aged 40–79 years who underwent nationwide health screening in 2009, since when high-density lipoprotein (HDL) cholesterol and creatinine levels have been measured (Fig 1). The health screening and NHIS reimbursement records were collected from 2006 to 2017. I excluded the following cases from these one million adults: 57,201 with missing or outlier data; 150,072 with a medical history of heart disease, stroke, or cancer or with an estimated glomerular filtration rate (eGFR) of <30 ml/min/1.73 m$^2$; and 83,877 who died or developed CVD, major cancers, or end-stage kidney disease (ESKD) before the baseline (i.e., January 1, 2012). From the remaining 708,850 adults, 61,665 users who had initiated statins during the 2007–2010 period and 601,494 nonusers who had never or rarely used statins before the baseline were identified. As past statin users might be more likely to have unmeasured health problems, the users who received statins in the year of cohort entry were excluded to minimize residual confounding. The analysis also excluded the users who initiated statins in the last year before baseline, since their statin use status could not reliably be assessed at the baseline. The propensity score (i.e., the predicted probability that a person would be a statin

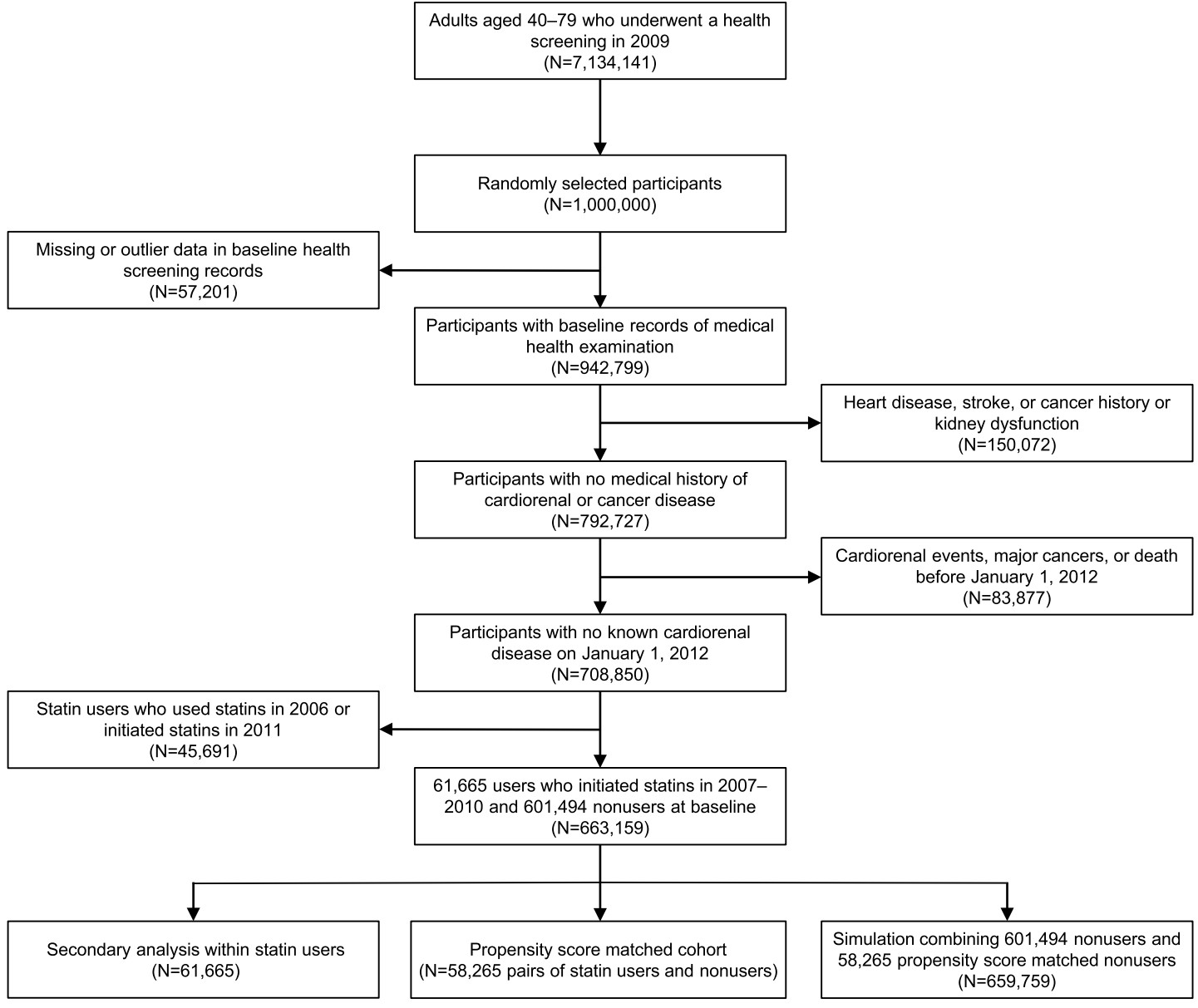

**Fig 1. Flow chart of participant selection.**

user) among 663,159 adults was calculated using a logistic regression with measured baseline covariates (i.e., age, sex, family history of CVD, income, physical exercise, alcohol consumption, smoking, body mass index, systolic blood pressure, fasting blood glucose, untreated total cholesterol, HDL cholesterol, proteinuria, and eGFR). Then, the pairs of statin users and nonusers were created requesting exact matches for the CVD risk categories and employing a greedy, nearest-neighbor matching algorithm with a caliper width of 0.2.

## Exposures and covariates

In each year of follow-up, statin use status (regular, intermittent, occasional, or nonuse; S1 Table in S1 File) was determined as the primary exposure of interest, using information on the

prescription of drugs (S2 Table in S1 File). Regular use was determined if statins were used for >2/3 of each period of statin therapy (from the initiation of medication to each year of follow-up), intermittent use if the drugs were used for 1/3–2/3 of each period, occasional use if the drugs were used for <1/3 of each period, and nonuse if the drugs were not used for ≥90 days per year. In each year, antihypertensive and antidiabetic use statuses (regular, irregular, or non-use) were also determined: regular or irregular use was determined if a case received these drugs for >1/2 or ≤1/2 of each period of medication, correspondingly. Meanwhile, non-use was determined if a case did not receive these drugs for ≥90 days per year.

Using the baseline data, I categorized age (40–44, 45–49, 50–54, 55–59, 60–64, 65–69, 70–74, or 75–79 years), sex (men or women), family history of CVD (yes or no), income (high, middle, or low), physical exercise (<1, 1–2, 3–4, or ≥5 days/week), alcohol consumption (0, 0.1–0.4, 0.5–1.4, 1.5–2.9, or ≥3.0 drinks/day), smoking (never, former, or current smoker), body mass index (10.0–18.4, 18.5–22.9, 23.0–24.9, 25.0–29.9, or 30.0–50 kg/m$^2$), systolic blood pressure (90–109, 110–119, 120–129, 130–139, 140–149, 150–159, or 160–200 mm Hg), fasting blood glucose (1.7–4.3, 4.4–5.5, 5.6–6.9, 7.0–7.7, 7.8–8.8, 8.9–9.9, 10.0–50.0 mmol/l), untreated total cholesterol (3.37–4.13, 4.14–5.17, 5.18–6.21, 6.22–7.24, or 7.25–8.29 mmol/l; for a missing value, I imputed the product of treated total cholesterol and 1.10 which was the mean ratio of untreated to treated levels in participants in whom both levels were available), HDL cholesterol (0.52–1.03, 1.04–1.54, or 1.55–2.59 mmol/l), proteinuria (yes or no), and eGFR (30–44, 45–59, 60–89, or ≥90 ml/min/1.73 m$^2$). Details on baseline covariates are provided in S1 Text and S3 Table in S1 File.

## Risk categories

Participants were assigned to one of three categories according to the 10-year risk for CVD (high,10.0–90.0%; moderate, 5.0–9.9%; or low, 0.1–4.9%) predicted using the 2018 revised Pooled Cohort Equations (S2 Text in S1 File) [16]. Participants were also categorized by the number of risk factors present at the baseline (≥3, 2, or ≤1 of the 5 risk factors—hypertension, diabetes, dyslipidemia, proteinuria, and active smoking; S1 Table in S1 File).

## Outcomes

The study outcomes were identified using NHIS reimbursement record-retrieved information on in-hospital procedures, surgeries, and medications (S2 Table in S1 File), along with the diagnosis codes in the NHID (S4 Table in S1 File). The primary outcomes were major CVD events and all-cause deaths by December 31, 2017. Major CVD event was identified as a composite of acute myocardial infarction, acute ischemic stroke, and cardiovascular death: i.e., revascularization or critical care unit admission for myocardial infarction, revascularization or critical care unit admission for ischemic stroke, and death from CVD. Vital status was confirmed using death certificates from Statistics Korea.

The secondary outcomes comprised five non-CVD events. Diabetes mellitus was identified as a fasting glucose of ≥7.0 mmol/l during biennial health screenings or a prescription of anti-diabetics for ≥90 days per year. Dementia cases were defined as those that received cholinesterase inhibitors or memantine for dementia: in Korea, the medications could be reimbursed in the cases with mini mental state exam ≤26, clinical dementia rating ≤3, or global deterioration scale ≤7. Severe cataract cases were defined as those that underwent first surgery for cataract: it comprised cataract surgery, primary intraocular lens implantation, or lens-phacoemulsification. ESKD was defined as dialysis for ≥90 days per year or kidney transplantation and identified using information about hemodialysis, prescribed peritoneal dialysates, and kidney transplantation. Major cancers comprised the seven most common causes of

cancer death in Korea: i.e., lung cancer, hepatoma, colon cancer, stomach cancer, pancreatic cancer, gallbladder and bile duct cancer, and breast cancer.

## Statistical analysis

The primary analyses compared the outcome risks in users who had recently initiated statins with those in nonusers who had never or rarely received statins at the baseline. To avoid selection bias due to loss of follow-up, baseline nonusers, including those who received statins thereafter, were used as references throughout the study period. In propensity score-matched 58,265 pairs, the hazard ratios were estimated using Cox regression with time-varying covariates to incorporate changes in medication use over time. The yearly updated statin, antihypertensive, and antidiabetic use status were entered as time-lagged covariates for subsequent years (S5 Table in S1 File) [17]. The categorized baseline covariates were entered as fixed covariates to address potential imbalance resulting from further classification of statin users according to statin use status [18]. The analyses were conducted for all participants combined and for subgroups stratified by age ($\geq$65 or <65 years), sex (men or women), or predicted CVD risk.

Secondary analyses were conducted to explore residual confounding related to statin indication. The hazard ratios of regular use were estimated in comparison with occasional rather than nonuse in a 61,665 users having initiated statins in the 2007–2010 period. The Cox models included the yearly medication use status as time-varying covariates and the baseline status along with the year of statin initiation as fixed covariates. To explore the robustness of the results, the hazard ratios were reanalyzed in a NHIS-generated sample cohort (S3 Text in S1 File). Further, to explore the validity of the revised Pooled Cohort Equations, the hazard ratios for major CVD events were estimated with the predicted 10-year risk for CVD using restricted cubic spline functions, in response to peer reviewer comments.

To estimate the absolute benefits of statins in the Korean population, simulation analyses were conducted combining the 601,494 baseline nonusers of the full cohort and the 58,265 pairs of the propensity score-matched cohort: the 58,265 nonusers of the matched cohort were double counted as simulated nonusers to replace 58,265 statin users. Statistical analyses were performed using SAS (version 9.4; SAS Institute, Cary, NC, USA) and SPSS (version 25.0; SPSS Inc, Chicago, IL, USA). Data are presented as numbers and percentages, means and SDs, or hazard ratios and 95% CIs.

## Results

### Baseline characteristics

The propensity score-matched cohort included 58,265 pairs of statin users and nonusers. They had baseline characteristics with standardized differences of mostly less than 0.10 (Table 1) [19]. Among statin users, the occasional, intermittent, and regular user groups had comparable demographic characteristics. The regular statin user group had higher proportions of regular antihypertensive and antidiabetic users than the other two groups.

### Statin use and outcome risks

In the propensity score matched cohort of 58,265 pairs of statin users and nonusers, major CVD events occurred in 1105 nonusers and 818 statin users, and all-cause deaths were noted in 2246 nonusers and 1797 users, over 6 years of follow-up. Multivariable-adjusted hazard ratios compared with baseline nonusers were estimated according to statin use status (Fig 2). The hazard ratios (95% CI) for major CVD events were 1.06 (0.93–1.20) in occasional users, 0.82 (0.70–0.97) in intermittent users, and 0.57 (0.50–0.64) in regular users. The

**Table 1. Baseline characteristics of the propensity score-matched cohort.**

| Characteristic | Baseline Statin Users | | | | Baseline Nonusers | Standardized Difference[a] |
|---|---|---|---|---|---|---|
| | Occasional Users | Intermittent Users | Regular Users | Total | | |
| No. of participants | 16085 | 11360 | 30820 | 58265 | 58265 | |
| 10-y CVD risk | | | | | | 0.000 |
| ≥10.0%, no. (%) | 9066 (56.4%) | 6118 (53.9%) | 14115 (45.8%) | 29299 (50.3%) | 29299 (50.3%) | |
| 5.0–9.9%, no. (%) | 3646 (22.7%) | 2643 (23.3%) | 7807 (25.3%) | 14096 (24.2%) | 14096 (24.2%) | |
| <5.0%, no. (%) | 3373 (21.0%) | 2599 (22.9%) | 8898 (28.9%) | 14870 (25.5%) | 14870 (25.5%) | |
| Age, mean (SD), y | 58.2 (9.1) | 58.9 (8.9) | 59.9 (9.0) | 59.2 (9.0) | 59.1 (9.0) | 0.014 |
| Male, no. (%) | 7439 (46.2%) | 4805 (42.3%) | 14015 (45.5%) | 26259 (45.1%) | 26774 (46.0%) | 0.018 |
| Family history of CVD, no. (%) | 1642 (10.2%) | 1165 (10.3%) | 3503 (11.4%) | 6310 (10.8%) | 5918 (10.2%) | 0.022 |
| Income level, no. (%) | | | | | | 0.026 |
| High | 3613 (22.5%) | 2509 (22.1%) | 7109 (23.1%) | 13231 (22.7%) | 12764 (21.9%) | |
| Middle | 6241 (38.8%) | 4300 (37.9%) | 11809 (38.3%) | 22350 (38.4%) | 22309 (38.3%) | |
| Low | 6231 (38.7%) | 4551 (40.1%) | 11902 (38.6%) | 22684 (38.9%) | 23192 (39.8%) | |
| Antihypertensive use, no. (%) | | | | | | 0.021 |
| Never use | 8627 (53.6%) | 5084 (44.8%) | 8925 (29.0%) | 22636 (38.9%) | 21970 (37.7%) | |
| Irregular use | 1071 (6.7%) | 576 (5.1%) | 558 (1.8%) | 2205 (3.8%) | 2220 (3.8%) | |
| Regular use | 6387 (39.7%) | 5700 (50.2%) | 21337 (69.2%) | 33424 (57.4%) | 34075 (58.5%) | |
| Antidiabetic use, no. (%) | | | | | | 0.024 |
| Never use | 13525 (84.1%) | 8908 (78.4%) | 21800 (70.7%) | 44233 (75.9%) | 44975 (77.2%) | |
| Irregular use | 358 (2.2%) | 191 (1.7%) | 170 (0.6%) | 719 (1.2%) | 718 (1.2%) | |
| Regular use | 2202 (13.7%) | 2261 (19.9%) | 8850 (28.7%) | 13313 (22.8%) | 12572 (21.6%) | |
| Physical exercise, no. (%) | | | | | | 0.038 |
| <1 day/week | 5430 (33.8%) | 3832 (33.7%) | 10659 (34.6%) | 19921 (34.2%) | 20143 (34.6%) | |
| 1–2 days/week | 6035 (37.5%) | 4034 (35.5%) | 10954 (35.5%) | 21023 (36.1%) | 21124 (36.3%) | |
| 3–4 days/week | 3289 (20.4%) | 2461 (21.7%) | 6531 (21.2%) | 12281 (21.1%) | 12100 (20.8%) | |
| ≥5 days/week | 1331 (8.3%) | 1033 (9.1%) | 2676 (8.7%) | 5040 (8.7%) | 4898 (8.4%) | |
| Smoking, no. (%) | | | | | | 0.080 |
| Never smoked | 10634 (66.1%) | 7781 (68.5%) | 20404 (66.2%) | 38819 (66.6%) | 37856 (65.0%) | |
| Former smoker | 2724 (16.9%) | 1800 (15.8%) | 5454 (17.7%) | 9978 (17.1%) | 9609 (16.5%) | |
| Current smoker | 2727 (17.0%) | 1779 (15.7%) | 4962 (16.1%) | 9468 (16.2%) | 10800 (18.5%) | |
| Alcohol consumption, no. (%) | | | | | | 0.045 |
| <0.1 drinks/day | 9045 (56.2%) | 6604 (58.1%) | 17633 (57.2%) | 33282 (57.1%) | 32958 (56.6%) | |
| 0.1–0.4 drinks/day | 1957 (12.2%) | 1359 (12.0%) | 3468 (11.3%) | 6784 (11.6%) | 6572 (11.3%) | |
| 0.5–1.4 drinks/day | 2109 (13.1%) | 1451 (12.8%) | 4095 (13.3%) | 7655 (13.1%) | 7622 (13.1%) | |
| 1.5–2.9 drinks/day | 1565 (9.7%) | 1019 (9.0%) | 2964 (9.6%) | 5548 (9.5%) | 5801 (10.0%) | |

*(Continued)*

**Table 1.** (Continued)

| Characteristic | Baseline Statin Users | | | | Baseline Nonusers | Standardized Difference[a] |
|---|---|---|---|---|---|---|
| | Occasional Users | Intermittent Users | Regular Users | Total | | |
| ≥3.0 drinks/day. | 1409 (8.8%) | 927 (8.2%) | 2660 (8.6%) | 4996 (8.6%) | 5312 (9.1%) | |
| Proteinuria, no. (%) | 1089 (6.8%) | 903 (7.9%) | 2831 (9.2%) | 4823 (8.3%) | 6119 (10.5%) | 0.076 |
| BMI, mean (SD), kg/m$^2$ | 24.7 (2.9) | 24.8 (2.9) | 25.2 (3.0) | 25.0 (2.9) | 24.9 (2.9) | 0.002 |
| SBP, mean (SD), mm Hg | 126.7 (12.6) | 127.7 (12.8) | 129.6 (12.6) | 128.5 (12.7) | 128.8 (12.6) | 0.025 |
| FBG, mean (SD), mmol/l | 5.78 (1.50) | 5.93 (1.64) | 6.14 (1.73) | 6.00 (1.66) | 6.03 (1.71) | 0.025 |
| Untreated total cholesterol, mean (SD), mmol/l | 5.81 (0.85) | 5.89 (0.91) | 5.71 (0.89) | 5.77 (0.89) | 5.65 (0.85) | 0.137 |
| HDL cholesterol, mean (SD), mmol/l | 1.39 (0.31) | 1.41 (0.31) | 1.39 (0.30) | 1.40 (0.31) | 1.39 (0.31) | 0.033 |
| eGFR, mean (SD) ml/min/1.73 m$^2$ | 83.5 (14.2) | 83.0 (14.2) | 82.1 (14.7) | 82.7 (14.5) | 82.7 (14.4) | 0.001 |

[a] The standardized differences were calculated to assess post-match balance in the baseline covariates between baseline statin users and nonusers.

BMI, body mass index; CVD, cardiovascular disease; eGFR, estimated glomerular filtration rate; FBG, fasting blood glucose; SBP, systolic blood pressure.

corresponding hazard ratios (95% CI) for all-cause mortality were 1.01 (0.92–1.10), 0.87 (0.78–0.98), and 0.71 (0.66–0.77), respectively.

In the secondary analyses that estimated the hazard ratios in comparison with occasional rather than nonusers, the risk reductions for major CVD events and all-cause mortality in regular statin users were substantial with the hazard ratios similar to those in the primary analyses (Fig 3).

For non-CVD events, regular statin use was associated with a risk increase for diabetes (hazard ratio 1.16, 95% CI 1.10–1.21) and a risk reduction for dementia (hazard ratio 0.88, 95% CI 0.79–0.98). The trends of the associations persisted in secondary analyses conducted within statin users and in further analyses conducted using the NHIS-generated sample cohort (S1 Fig in S1 File). However, statin use was not consistently associated with the risk for major cancers, ESKD, or cataract surgery.

In stratified analyses, the hazard ratios for major CVD events and all-cause mortality in statin users were similar, irrespective of age, sex, or predicted CVD risk (Fig 4). When the absolute event rates were estimated in statin users with reference to the rates in nonusers, the absolute risk reductions for major CVD events and all-cause mortality were greater among higher risk categories (S2 Fig in S1 File).

## Simulation analyses

Simulation was performed on the assumption that the relative risk reductions of 43.3% for major CVD events and 28.8% for all-cause mortality were not different between the simulated and propensity score-matched cohorts, nor did they vary among the CVD risk categories. When 659,759 simulated pairs were categorized using the Pooled Cohort Equations, the major CVD events were observed or estimated to occur in 3580 nonusers and 2030 regular users among the 90,750 pairs with a 10-year risk of ≥10.0% (Table 2), over 6 years of follow-up. When categorized by the number of risk factors, major CVD events occurred or would occur in 3640 nonusers and 2064 regular users among the 161,071 pairs with ≥2 risk factors. Thus, over the 6-year follow-up, regular statin use could reduce 1550 CVDs and 2573 deaths in 90,750 adults (17 CVDs and 28 deaths in 1000) with a risk of ≥10.0% (Fig 5). Meanwhile, regular use could reduce 1576 CVDs and 2254 deaths in 161,071 adults (10 CVDs and 14 deaths in 1000) with ≥2 risk factors. However, in the actual cohort of adults with a risk of ≥10%, statins

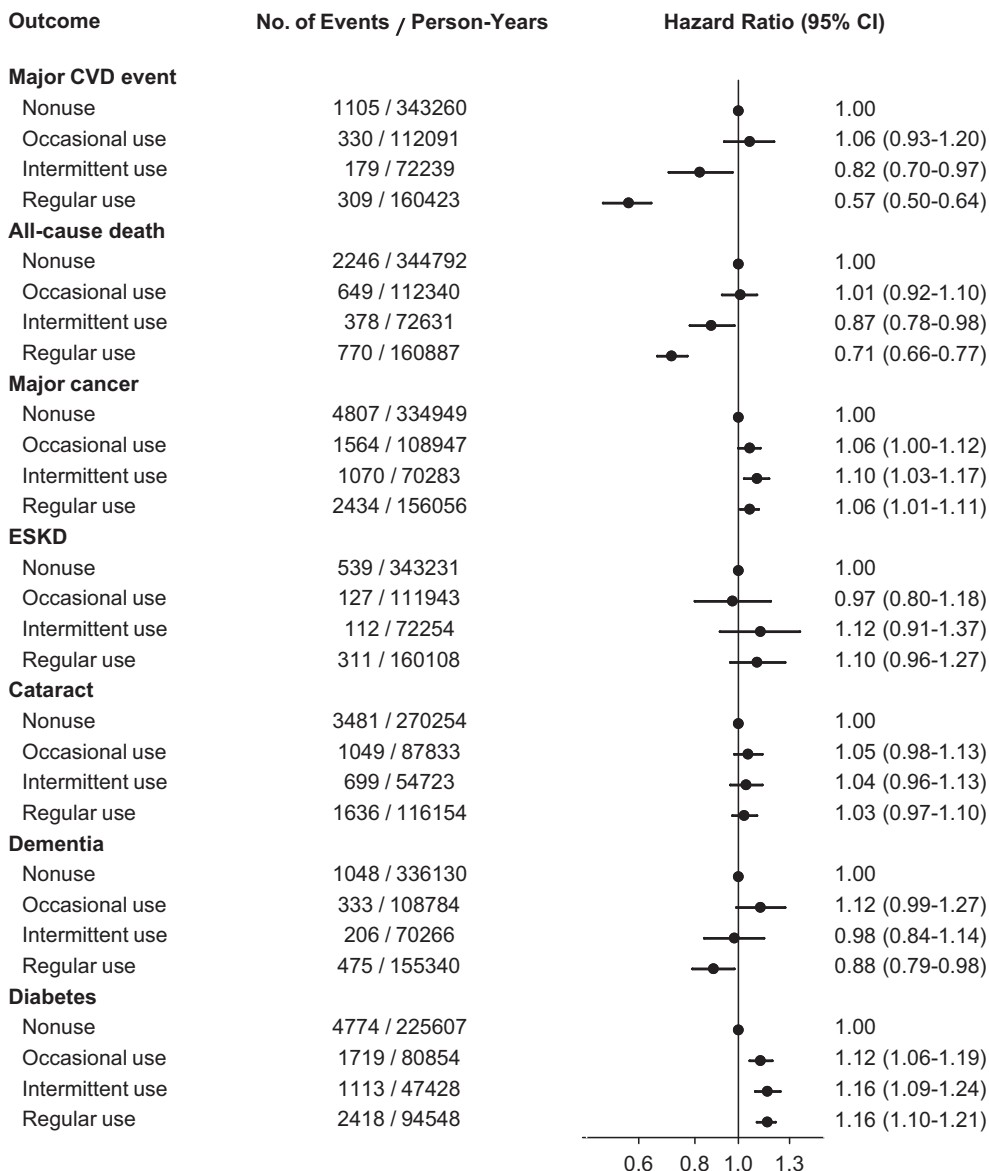

| Outcome | No. of Events / Person-Years | Hazard Ratio (95% CI) |
|---|---|---|
| **Major CVD event** | | |
| Nonuse | 1105 / 343260 | 1.00 |
| Occasional use | 330 / 112091 | 1.06 (0.93-1.20) |
| Intermittent use | 179 / 72239 | 0.82 (0.70-0.97) |
| Regular use | 309 / 160423 | 0.57 (0.50-0.64) |
| **All-cause death** | | |
| Nonuse | 2246 / 344792 | 1.00 |
| Occasional use | 649 / 112340 | 1.01 (0.92-1.10) |
| Intermittent use | 378 / 72631 | 0.87 (0.78-0.98) |
| Regular use | 770 / 160887 | 0.71 (0.66-0.77) |
| **Major cancer** | | |
| Nonuse | 4807 / 334949 | 1.00 |
| Occasional use | 1564 / 108947 | 1.06 (1.00-1.12) |
| Intermittent use | 1070 / 70283 | 1.10 (1.03-1.17) |
| Regular use | 2434 / 156056 | 1.06 (1.01-1.11) |
| **ESKD** | | |
| Nonuse | 539 / 343231 | 1.00 |
| Occasional use | 127 / 111943 | 0.97 (0.80-1.18) |
| Intermittent use | 112 / 72254 | 1.12 (0.91-1.37) |
| Regular use | 311 / 160108 | 1.10 (0.96-1.27) |
| **Cataract** | | |
| Nonuse | 3481 / 270254 | 1.00 |
| Occasional use | 1049 / 87833 | 1.05 (0.98-1.13) |
| Intermittent use | 699 / 54723 | 1.04 (0.96-1.13) |
| Regular use | 1636 / 116154 | 1.03 (0.97-1.10) |
| **Dementia** | | |
| Nonuse | 1048 / 336130 | 1.00 |
| Occasional use | 333 / 108784 | 1.12 (0.99-1.27) |
| Intermittent use | 206 / 70266 | 0.98 (0.84-1.14) |
| Regular use | 475 / 155340 | 0.88 (0.79-0.98) |
| **Diabetes** | | |
| Nonuse | 4774 / 225607 | 1.00 |
| Occasional use | 1719 / 80854 | 1.12 (1.06-1.19) |
| Intermittent use | 1113 / 47428 | 1.16 (1.09-1.24) |
| Regular use | 2418 / 94548 | 1.16 (1.10-1.21) |

0.6  0.8  1.0  1.3

**Fig 2. Hazard ratios for outcomes according to statin use status in the propensity score-matched cohort.** Among 58,265 pairs matched on propensity scores, multivariable-adjusted hazard ratios were estimated using Cox models with a time-varying covariate for statin use status. Baseline nonusers served as the reference. The participants who were diagnosed with cataract (or dementia, or diabetes) before the baseline were excluded from the analysis of cataract (or dementia, or diabetes). CVD, cardiovascular disease; ESKD, end-stage kidney disease.

were used in 29,299 adults (including occasional, intermittent, and regular users) and estimated to reduce 232 CVDs and 346 deaths (3 CVDs and 4 deaths in 1000).

## Discussion

In this cohort study of Korean adults with no prior CVD, regular statin use was associated with substantial risk reductions for major CVD events and all-cause mortality. For non-CVD events, the effects of statins were neutral. In stratified analyses, statin use was associated with similar relative risk reductions for major CVD events and all-cause mortality, irrespective of

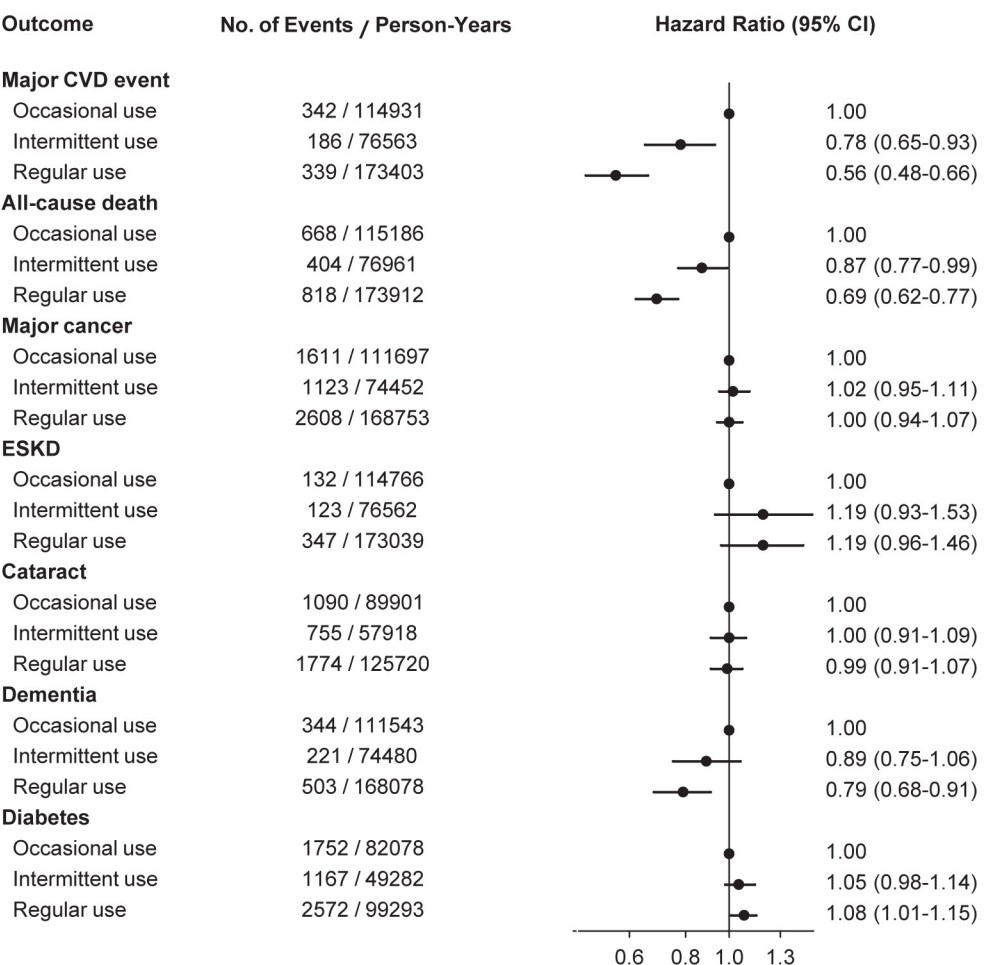

| Outcome | No. of Events / Person-Years | Hazard Ratio (95% CI) |
|---|---|---|
| **Major CVD event** | | |
| Occasional use | 342 / 114931 | 1.00 |
| Intermittent use | 186 / 76563 | 0.78 (0.65-0.93) |
| Regular use | 339 / 173403 | 0.56 (0.48-0.66) |
| **All-cause death** | | |
| Occasional use | 668 / 115186 | 1.00 |
| Intermittent use | 404 / 76961 | 0.87 (0.77-0.99) |
| Regular use | 818 / 173912 | 0.69 (0.62-0.77) |
| **Major cancer** | | |
| Occasional use | 1611 / 111697 | 1.00 |
| Intermittent use | 1123 / 74452 | 1.02 (0.95-1.11) |
| Regular use | 2608 / 168753 | 1.00 (0.94-1.07) |
| **ESKD** | | |
| Occasional use | 132 / 114766 | 1.00 |
| Intermittent use | 123 / 76562 | 1.19 (0.93-1.53) |
| Regular use | 347 / 173039 | 1.19 (0.96-1.46) |
| **Cataract** | | |
| Occasional use | 1090 / 89901 | 1.00 |
| Intermittent use | 755 / 57918 | 1.00 (0.91-1.09) |
| Regular use | 1774 / 125720 | 0.99 (0.91-1.07) |
| **Dementia** | | |
| Occasional use | 344 / 111543 | 1.00 |
| Intermittent use | 221 / 74480 | 0.89 (0.75-1.06) |
| Regular use | 503 / 168078 | 0.79 (0.68-0.91) |
| **Diabetes** | | |
| Occasional use | 1752 / 82078 | 1.00 |
| Intermittent use | 1167 / 49282 | 1.05 (0.98-1.14) |
| Regular use | 2572 / 99293 | 1.08 (1.01-1.15) |

**Fig 3. Hazard ratios for outcomes according to statin use status within baseline statin users.** Within 61,665 statin users, multivariable-adjusted hazard ratios were estimated using Cox models with a time-varying covariate for statin use status. Occasional users served as the reference. The participants who were diagnosed with cataract (or dementia, or diabetes) before the baseline were excluded from the analysis of cataract (or dementia, or diabetes). CVD, cardiovascular disease; ESKD, end-stage kidney disease.

age, sex, or predicted CVD risk. Accordingly, the absolute risk reductions with statins were substantially greater in adults at higher risk predicted using CVD risk calculators. However, in this cohort, statin use was insufficient and estimated to reduce much smaller numbers of events among adults at high predicted risk compared with those in simulation analyses.

In both the primary and secondary analyses, regular statin use was associated with substantial risk reductions of CVD events and all-cause mortality. The 43% risk reduction for major CVD events in regular users vs nonusers was similar to the 44% risk reduction for those in the secondary analysis (i.e., in regular vs occasional users) and comparable to the 24%–44% CVD risk reductions observed in primary prevention statin trials [2–8]. Conversely, regular statin use was associated with a 16% risk increase for incident diabetes in the primary analysis (or with an 8% risk increase in the secondary analysis). In 2010 and 2016 meta-analyses of clinical trials, no or only minimal associations were found between statin use and diabetes risk [2,20]. However, some cohort studies reported that statin use was associated with 50%–100% risk increase for diabetes compared with nonuse [21–24]. Unlike previous cohort studies, the

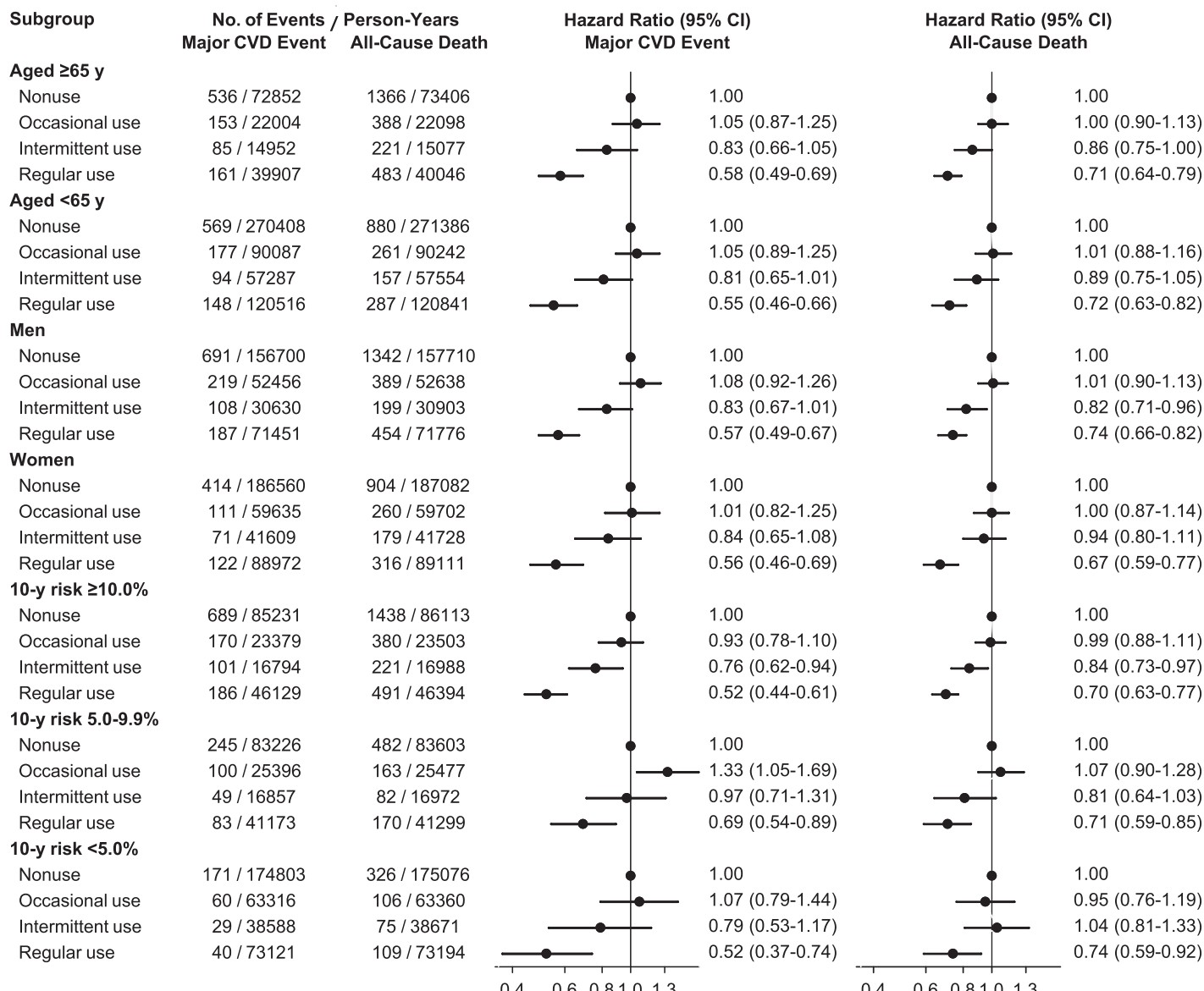

**Fig 4. Hazard ratios for primary outcomes in stratified analyses.** Multivariable-adjusted hazard ratios were estimated using Cox models with a time-varying covariate for statin use status, in subgroups stratified by age, sex, or 10-year CVD risk. CVD, cardiovascular disease.

present study generated the propensity score-matched cohort among users having recently initiated statins and a relatively large number of nonusers. Furthermore, the secondary analysis compared regular use with occasional rather than nonuse to assess residual confounding related to statin indication. The excess risk of diabetes observed in previous cohort studies of statin users might be exaggerated due to confounding by indication or reverse causality, i.e. participants with impending diabetes would more likely receive statins. Meanwhile, regular statin use was associated with a risk reduction for dementia in this study. Given substantial CVD and mortality benefits of regular use, statins should not be discontinued due to exaggerated fear of non-CVD side effects [25].

In stratified analyses, regular statin use had an association with similar relative risk reductions for CVD events and all-cause mortality, irrespective of age, sex, or predicted CVD risk.

**Table 2. Baseline characteristics and follow-up outcomes in the simulation cohort.**

| Characteristic | 10-y CVD Risk | | | |
|---|---|---|---|---|
| | ≥10.0% | 5.0–9.9% | <5.0% | Total |
| No. of pairs | 90750 | 119013 | 449996 | 659759 |
| Baseline characteristic | | | | |
| Age, mean (SD), y | 65.5 (8.9) | 58.7 (9.3) | 50.8 (6.9) | 54.2 (9.4) |
| Men, no. (%) | 77119 (85.0%) | 93591 (78.6%) | 176089 (39.1%) | 346799 (52.6%) |
| SBP, mean (SD), mm Hg | 134.7 (13.1) | 128.7 (11.4) | 120.2 (11.4) | 123.7 (12.9) |
| Untreated total cholesterol, mean (SD), mmol/l | 5.21 (0.80) | 5.25 (0.79) | 5.12 (0.76) | 5.15 (0.77) |
| HDL cholesterol, mean (SD), mmol/l | 1.27 (0.29) | 1.33 (0.29) | 1.46 (0.31) | 1.41 (0.31) |
| Diabetes, no. (%) | 37729 (41.6%) | 16547 (13.9%) | 12582 (2.8%) | 66858 (10.1%) |
| Smoking, no. (%) | 45502 (50.1%) | 50158 (42.1%) | 51040 (11.3%) | 146700 (22.2%) |
| Antihypertensive use, no. (%) | 44050 (48.5%) | 37093 (31.2%) | 59847 (13.3%) | 140990 (21.4%) |
| Major CVD event, no. (%) | | | | |
| Simulated nonusers[a] | 3580 (8.8%) | 1799 (1.5%) | 1530 (0.3%) | 6909 (1.0%) |
| Simulated regular users[b] | 2030 (5.0%) | 1020 (0.8%) | 868 (0.2%) | 3917 (0.6%) |
| Actual cohort[c] | 3348 (8.2%) | 1786 (1.5%) | 1488 (0.3%) | 6622 (1.0%) |
| All-cause death, no. (%) | | | | |
| Simulated nonusers[a] | 8935 (21.9%) | 3939 (3.3%) | 3827 (0.8%) | 16701 (2.5%) |
| Simulated regular users[b] | 6362 (15.6%) | 2805 (2.3%) | 2725 (0.5%) | 11891 (1.8%) |
| Actual cohort[c] | 8589 (21.1%) | 3872 (3.2%) | 3791 (0.8%) | 16252 (2.5%) |
| Characteristic | Presence of Risk Factors | | | |
| | ≥3 Risk Factors | 2 Risk Factors | ≤1 Risk Factor | Total |
| No. of pairs | 40712 | 120359 | 498688 | 659759 |
| Baseline characteristic | | | | |
| Age, mean (SD), y | 57.5 (9.8) | 56.6 (9.9) | 53.4 (9.1) | 54.2 (9.4) |
| Men, no. (%) | 32642 (80.2%) | 82988 (69.0%) | 231169 (46.4%) | 346799 (52.6%) |
| Hypertension, no. (%) | 35627 (87.5%) | 80835 (67.2%) | 91267 (18.3%) | 207729 (31.5%) |
| Dyslipidemia, no. (%) | 30370 (74.6%) | 35213 (29.3%) | 88822 (17.8%) | 154405 (23.4%) |
| Proteinuria, no. (%) | 13540 (33.3%) | 12983 (10.8%) | 9699 (1.9%) | 36222 (5.5%) |
| Diabetes, no. (%) | 25656 (63.0%) | 29481 (24.5%) | 11721 (2.4%) | 66858 (10.1%) |
| Smoking, no. (%) | 25223 (62.0%) | 53873 (44.8%) | 67604 (13.6%) | 146700 (22.2%) |
| Major CVD event, no. (%) | | | | |
| Simulated nonusers[a] | 1428 (3.5%) | 2212 (1.8%) | 3269 (0.7%) | 6909 (1.0%) |
| Simulated regular users[b] | 810 (2.0%) | 1254 (1.0%) | 1854 (0.4%) | 3917 (0.6%) |
| Actual cohort[c] | 1276 (3.1%) | 2102 (1.7%) | 3244 (0.7%) | 6622 (1.0%) |
| All-cause death, no. (%) | | | | |
| Simulated nonusers[a] | 2752 (6.8%) | 5072 (4.2%) | 8877 (1.8%) | 16701 (2.5%) |
| Simulated regular users[b] | 1959 (4.8%) | 3611 (3.0%) | 6320 (1.3%) | 11891 (1.8%) |
| Actual cohort[c] | 2533 (6.2%) | 4861 (4.0%) | 8858 (1.8%) | 16252 (2.5%) |

[a] The values were the numbers of events observed in simulated nonusers over 6 years of follow-up.

[b] The numbers of events in simulated regular statin users were estimated on the assumption that the relative risk reductions of 43.3% for CVD and 28.8% for mortality were not different between the simulated and propensity score-matched cohorts, nor were they different among the CVD risk categories.

[c] The values were the numbers of events observed in the actual cohort combining 601,494 baseline nonusers and 58,265 statin users over 6 years of follow-up.

CVD, cardiovascular disease; SBP, systolic blood pressure.

The findings were consistent with those of statin trials indicating similarities among relative risk reductions for CVD events across subgroups stratified by baseline characteristics or predicted CVD risk [7,8,26]. Meanwhile, in the present study, regular statin use was not

**Population-Based Cohort of 659,759 Adults Aged 40 to 79 Years**

**Fig 5. Simulation for the Korean population with no prior CVD.** The numbers of events in simulated regular statin users were estimated on the assumption that the relative risk reductions of 43.3% for CVD and 28.8% for mortality were not different between the simulated and propensity score-matched cohorts, nor were they different among the CVD risk categories. CVD, cardiovascular disease.

consistently associated with the risk for major cancers, ESKD, or cataract, to which elderly or high risk individuals might be more vulnerable. The relative risk reductions for CVD events were steady, and the effects on non-CVD events were neutral. Therefore, the absolute net benefits of statins were substantially greater among participants at higher risk predicted using CVD risk calculators that accounted for age, sex, and other risk factors. By comparison, a study for antihypertensive users showed that aggressive lowering of blood pressure was associated with a risk increase for adverse outcomes. It also concluded that the risk calculators failed to discriminate the risk groups according to different risk thresholds of treated blood pressure [27]. The present findings strongly support the validity of CVD risk calculators in determining potential beneficiaries of statin use. However, this study does not ensure the role of risk calculators in determining optimal blood pressure for antihypertensive treatment.

In this study, regular statin use was associated with relative risk reductions for CVD events even among individuals at low predicted risk. However, the small absolute benefits could not support the need of primary prevention statin therapy in low risk individuals (S2 Fig in S1 File). In contrast, statins would certainly be beneficial in individuals at high risk. When simulation was performed for the population-based cohort of Korean adults, the absolute risk reductions with statins in the population with a 10-year risk of ≥10% were similar to those in the population with ≥2 risk factors. The similarity existed although the population size ratio of the former to the latter was approximately 50%. This indicates the 10-year risk predicted using CVD risk calculators is more cost-effective in decision-making for statin therapy than the risk assessment by counting risk factors. However, statin use was insufficient and estimated to reduce much smaller numbers of CVDs and deaths among actual adults with a risk of ≥10%. Calculating 10-year CVD risk and regular statin use in individuals at high predicted risk could be an effective strategy to reduce outcome risks much more than present in Korea.

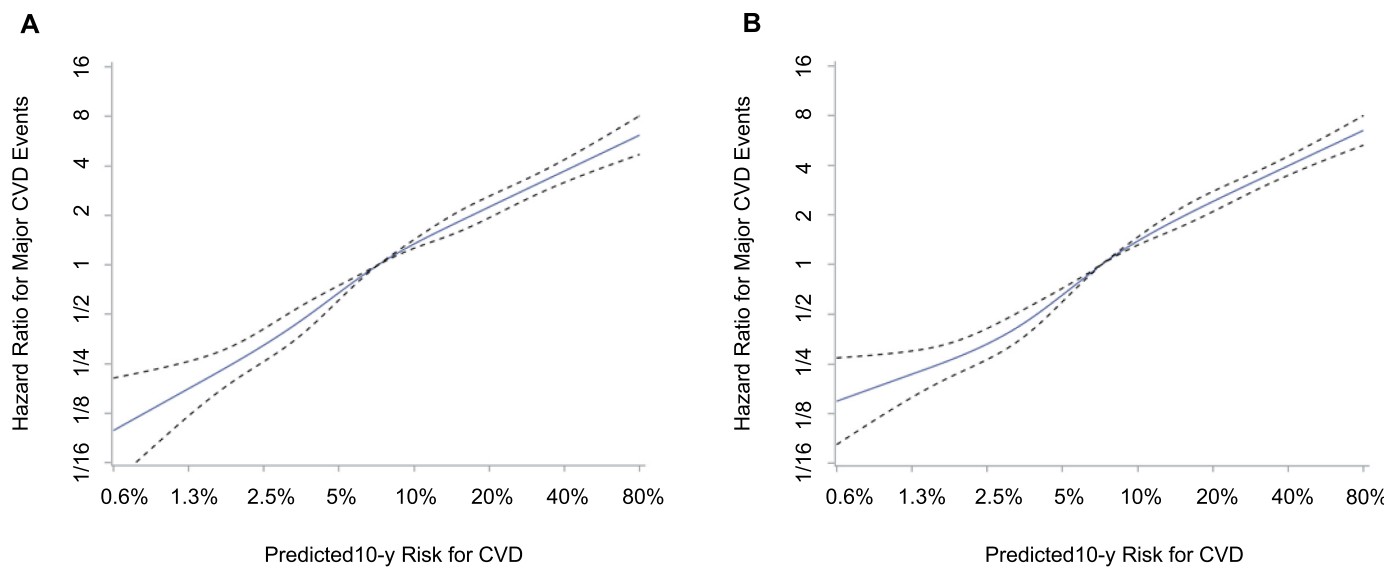

**Fig 6. Hazard ratios for CVD according to predicted risk scores.** Among propensity score-matched statin users (A) and nonusers (B), unadjusted hazard ratios (solid line) and 95% CIs (dotted line) were estimated using a restricted cubic spline function with five knots placed at the 5th, 25th, 50th, 75th, and 95th percentile CVD risk scores. CVD, cardiovascular disease.

The Pooled Cohort Equations risk calculator was designed to predict 10-year risk for fatal and non-fatal myocardial infarction and stroke that could be reduced by statin therapy. The calculator was developed and validated in several US cohorts comprising Caucasians and African Americans [16,28]. In this Korean cohort, the predicted risk scores were linearly associated with the hazard ratios for CVD events reflecting consistent discrimination over the risk scores (Fig 6), although the finding did not ensure individual-level accuracy or exclude a need of calibration to predict absolute risk. The use of the risk calculator to determine appropriate candidates for statin therapy, who will have a substantial net benefit with minimal harm or discomfort, requires additional study in both the US and other countries. Of the participants at intermediate risk predicted using the Pooled Cohort Equations, approximately 50% had no coronary artery calcification on computed tomography scanning [29], in an analysis of the Multi-Ethnic Study of Atherosclerosis. Moreover, CVD events were scarcely observed in those with no coronary artery calcification [29]. If statins were allocated to such patients, a large number of patients would be treated for a long time to prevent one CVD event. Accordingly, it would lead to many patients taking medicine daily, visiting medical offices periodically, and being exposed to potential side effects, while the CVD benefit of absolute risk reduction would be very small. Consideration of additional measures like coronary calcification might be needed for patients at intermediate predicted risk.

This study encountered several limitations that need addressing. First, despite the propensity score-matching, there might be residual confounding by unmeasured factors or inadequate matching. Specifically, this analysis did not address performance status or hospital visits that could potentially be related to regular use of statins, although the variables of physical exercise and medications use status were included in the models. Second, it did not evaluate muscle pain or damage, or liver injury, which might have a potential link with statin use. Moreover, the reasons for statin discontinuation (i.e., occasional or intermittent use) were not provided due to lack of the records in the NHID. Third, medication use status was assessed based on prescription data, and the analysis did not examine dose-dependent or drug-specific relationships of statins with outcome risks as doses of different statins varied with different

periods of time. Finally, caution is required when generalizing the findings to individuals with very old age or other ethnic groups. The reason was that the study included 40 to 79-year-old adults residing in Korea. Because the accuracy or validity of CVD risk calculators could be different according to race or ethnicity [30–32], the findings should be confirmed in other populations.

## Conclusions

The findings highlight the importance of regular statin use and strongly support the validity of CVD risk calculators in determining the potential benefits of statin use in Asian populations. The proper CVD risk assessment and regular statin use in patients at high predicted risk should be emphasized as an effective strategy for public health. Nevertheless, additional deliberation may be needed for patients at intermediate predicted risk, and further research is required to confirm the findings in populations with very old age or other ethnic groups.

## Supporting information

**S1 Checklist STROBE statement—checklist of items that should be included in reports of cohort studies.**
(DOCX)

**S1 File.**
(DOCX)

## Author Contributions

**Conceptualization:** Hae Hyuk Jung.

**Data curation:** Hae Hyuk Jung.

**Formal analysis:** Hae Hyuk Jung.

**Investigation:** Hae Hyuk Jung.

**Methodology:** Hae Hyuk Jung.

**Project administration:** Hae Hyuk Jung.

**Validation:** Hae Hyuk Jung.

**Visualization:** Hae Hyuk Jung.

**Writing – original draft:** Hae Hyuk Jung.

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
