## [Decision Letter · Decision Letter 0]

17 Dec 2020

PONE-D-20-35305

Statin use and outcome risks according to predicted CVD risk in Korea: A retrospective cohort study

PLOS ONE

Dear Dr. Jung,

Thank you for submitting your manuscript to PLOS ONE. After careful consideration, we feel that it has merit but does not fully meet PLOS ONE’s publication criteria as it currently stands. Therefore, we invite you to submit a revised version of the manuscript that addresses the points raised during the review process.

The editor and reviewers ahve read your manuscript and find it interesting and well-written. However there are come concerns raised by the reveiwers.  Please check them carefully and reoply clearly.

We look forward to receiving your revised manuscript.

Kind regards,

Katriina Aalto-Setala, Professor

Academic Editor

PLOS ONE

Journal Requirements:

Reviewers' comments:

Reviewer's Responses to Questions

**Comments to the Author**

1. Is the manuscript technically sound, and do the data support the conclusions?

Reviewer #1: Yes

Reviewer #2: Yes

2. Has the statistical analysis been performed appropriately and rigorously? 

Reviewer #1: Yes

Reviewer #2: I Don't Know

3. Have the authors made all data underlying the findings in their manuscript fully available?

Reviewer #1: Yes

Reviewer #2: Yes

4. Is the manuscript presented in an intelligible fashion and written in standard English?

Reviewer #1: Yes

Reviewer #2: Yes

5. Review Comments to the Author

Reviewer #1: This is a very interesting manuscript reporting a retrospective cohort study carried out in Korea, which addresses the efficacy of statin therapy in reducing major CVD events and all-cause deaths in primary prevention. The paper has according to my understanding initially been submitted to PloS Medicine, reviewed, revised and resubmitted, and then apparently transferred to PloS ONE. The original PloS Medicine review reports and the author’s rebuttal have been made available to me. The 3 reviewers of PloS Medicine have done a thorough job, concerning both the statistics employed and the actual rationale and design of the study.

The authors have, according to the original reviewers’ requirement, listed limitations of the study, one of which is that the type or dosage of statin was in the study not used for stratifying the data. The study population has been divided into non-users, occasional, intermittent and regular users, based on statin prescription records, so there is quite some uncertainty of what the factual use of the medication has been.

Major findings: Based on Korean public registers, major CVD event rates and all-cause mortality in 58,265 users who initiated statins during 2007–2010 were compared with those in 58,265 nonusers matched on propensity scores, from January 1, 2012 through December 31, 2017. Additionally, simulation was performed for a population based cohort of 659,759 adults. The CVD hazard ratios (95% CIs) in occasional, intermittent, and regular statin users were 1.06 (0.93–1.20), 0.82 (0.70–0.97), and 0.57 (0.50–0.64), respectively (43% risk reduction in regular users). The corresponding mortality hazard ratios were 1.01 (0.92–1.10), 0.87 (0.78–0.98), and 0.71 (0.66–0.77), respectively (29% risk reduction in regular statin users). In stratified analyses, the relative risk reductions were similar irrespective of age, sex, or predicted CVD risk. Thus, the absolute risk reductions were greater in higher risk categories. In simulation cohorts, regular statin use could reduce 1550 CVDs and 2573 deaths in 90,750 adults with a 10-year risk of ≥10.0%. However, in actual subjects with a risk of ≥10%, statin use was apparently insufficient since it was only estimated to reduce 232 CVDs and 346 deaths. The authors’ conclusion is that the study highlights the importance of regular statin use and supports the validity of CVD risk calculators in determining the potential benefits of statin use.

GENERAL COMMENT

In general, the work is adequately documented and the paper is well written, and after the careful revision according to the PloS Medicine reviewers’ comments, I see no major flaws in the manuscript. However, I wonder if the meticulous exclusion processes employed to ’distil’ relatively healthy statin users in the cohort may have, for a cryptic reason not detected by the authors, produced a subcohort with a reduced number of CVD events and deaths as compared to the respective non-users (for other reasons than the statin use). Have the authors given this possibility a careful thought? It would be good if the authors elaborated on this in the Discussion.

MINOR

On p. 13, the authors compare the SIMULATED CVDs (1550) and deaths (2573) in 90,750 adults with a risk equal of higher than 10%, to subjects in the actual cohort at the same risk level (29,299 subjects): 232 CVDs and 346 deaths. Since the sizes of the populations are very different (approx. 3x difference), comparison of the CVD and death numbers would be more concrete if the authors additionally indicated the % difference between the simulated and actual risks, also in the Abstract of the paper.

Reviewer #2: The author examined in retrospective cohort study the use of statins in primary prevention in Korean population scored for CVD risk and the CVD event rate. The study conclusion is that statins are most effective for patients at high CVD risk category. The study has high number of participants and the results are important for prevention of CVD in Asian populations effectively.

General comments:

The author should make a new literature research on statins in Asian population. He states at the beginning of introduction that “Statins are widely used to prevent cardiovascular disease (CVD) events: many studies have shown that statins reduce CVD-related morbidity and mortality in patients with or without prior CVD [1–3]. However, the questions about their harmful effects and beneficiaries, particularly in Asian populations, have remained unresolved.” In general the harmful effects and beneficiaries of statins is well studied. In Asian populations you can also find quite much evidence. This should be clarified here.

In conclusion section (in main manuscript and in abstract) should include the specific population of the study (Asians).

Specific comments:

Page 6 and other pages and tables: “Never use statins” group should be named as “non-users” in all section since the group includes patients not using statins over 90 days/year, but not never. Non-users term is used in result section, and I assume it means the same group. If this is not the case please clarify.

Page 7. Body mass index group 10.0-18.5. Are there patients having BMI 10.0-16? I would group these extremely anorectic individuals in different group (or exclude). At least some comment on this would be needed.

Page 7: dinking means alcohol consumption?

In discussion section, please specify if you talk about CVD mortality or all cause mortality.

Page 15. “Recent meta-analyses of clinical trials found no or only minimal associations between statin use and diabetes risk [2,20].” Recent is not 10 years ago.

Page 16. “However, they disapproved of the arguable role of risk calculators in determining optimal blood pressure for antihypertensive treatment.” I would recommend removing this sentence, it is not clear that you are referring other studies of yours here. If you like to keep this ad reference and clarify, maybe change the place for this sentence before you state that “Overall, these findings strongly support the validity of CVD risk calculators indetermining potential beneficiaries of statin use.” And then just use, “In this study, these findings…” And dont use "they" if you mean you. Take credits of your own work!

6. PLOS authors have the option to publish the peer review history of their article (what does this mean?). If published, this will include your full peer review and any attached files.

Reviewer #1: No

Reviewer #2: No

---

## [Author Response · Author response to Decision Letter 0]

23 Dec 2020

Manuscript ID: PONE-D-20-00850

I thank you for your thoughtful comments and suggestions. The manuscript has benefited from the comments. 

Response: I have confirmed that the manuscript meets PLOS ONE’s requirements. 

Response: The author cannot legally distribute the data. The data contain potentially identifying or sensitive patient information. The National Health Information Database can be requested from the National Health Insurance Sharing Service (http://nhiss.nhis.or.kr) and are provided in a “Data analysis room” located within the National Health Insurance Service (https://nhiss.nhis.or.kr/bd/ab/bdaba032eng.do). 

Response: N/A 

Response: I have included captions for the Supporting Information files at the end of the manuscript. I have checked in-text citations. 

Reviewer #1: This is a very interesting manuscript reporting a retrospective cohort study carried out in Korea, which addresses the efficacy of statin therapy in reducing major CVD events and all-cause deaths in primary prevention. The paper has according to my understanding initially been submitted to PloS Medicine, reviewed, revised and resubmitted, and then apparently transferred to PloS ONE. The original PloS Medicine review reports and the author’s rebuttal have been made available to me. The 3 reviewers of PloS Medicine have done a thorough job, concerning both the statistics employed and the actual rationale and design of the study.

The authors have, according to the original reviewers’ requirement, listed limitations of the study, one of which is that the type or dosage of statin was in the study not used for stratifying the data. The study population has been divided into non-users, occasional, intermittent and regular users, based on statin prescription records, so there is quite some uncertainty of what the factual use of the medication has been.

Major findings: Based on Korean public registers, major CVD event rates and all-cause mortality in 58,265 users who initiated statins during 2007–2010 were compared with those in 58,265 nonusers matched on propensity scores, from January 1, 2012 through December 31, 2017. Additionally, simulation was performed for a population based cohort of 659,759 adults. The CVD hazard ratios (95% CIs) in occasional, intermittent, and regular statin users were 1.06 (0.93–1.20), 0.82 (0.70–0.97), and 0.57 (0.50–0.64), respectively (43% risk reduction in regular users). The corresponding mortality hazard ratios were 1.01 (0.92–1.10), 0.87 (0.78–0.98), and 0.71 (0.66–0.77), respectively (29% risk reduction in regular statin users). In stratified analyses, the relative risk reductions were similar irrespective of age, sex, or predicted CVD risk. Thus, the absolute risk reductions were greater in higher risk categories. In simulation cohorts, regular statin use could reduce 1550 CVDs and 2573 deaths in 90,750 adults with a 10-year risk of ≥10.0%. However, in actual subjects with a risk of ≥10%, statin use was apparently insufficient since it was only estimated to reduce 232 CVDs and 346 deaths. The authors’ conclusion is that the study highlights the importance of regular statin use and supports the validity of CVD risk calculators in determining the potential benefits of statin use.

GENERAL COMMENT

In general, the work is adequately documented and the paper is well written, and after the careful revision according to the PloS Medicine reviewers’ comments, I see no major flaws in the manuscript. However, I wonder if the meticulous exclusion processes employed to ’distil’ relatively healthy statin users in the cohort may have, for a cryptic reason not detected by the authors, produced a subcohort with a reduced number of CVD events and deaths as compared to the respective non-users (for other reasons than the statin use). Have the authors given this possibility a careful thought? It would be good if the authors elaborated on this in the Discussion. 

Response: Thank you for your comments. As you pointed out, despite the exclusion of past statin users and the matching with propensity scores, there might be residual confounding by unmeasured factors or inadequate matching. Specifically, this analysis did not address performance status or hospital visits that could be related to regular use of statins, although the variables of physical activity and medications use status were included in the analytic models. I have revised the Limitations paragraph to include this limitation. 

MINOR

On p. 13, the authors compare the SIMULATED CVDs (1550) and deaths (2573) in 90,750 adults with a risk equal of higher than 10%, to subjects in the actual cohort at the same risk level (29,299 subjects): 232 CVDs and 346 deaths. Since the sizes of the populations are very different (approx. 3x difference), comparison of the CVD and death numbers would be more concrete if the authors additionally indicated the % difference between the simulated and actual risks, also in the Abstract of the paper.

Response: The Results and the Abstract sections have been revised to provide the differences as risk difference per 1000. Thank you for your reasonable recommendation for the manuscript.

Reviewer #2: The author examined in retrospective cohort study the use of statins in primary prevention in Korean population scored for CVD risk and the CVD event rate. The study conclusion is that statins are most effective for patients at high CVD risk category. The study has high number of participants and the results are important for prevention of CVD in Asian populations effectively.

General comments:

The author should make a new literature research on statins in Asian population. He states at the beginning of introduction that “Statins are widely used to prevent cardiovascular disease (CVD) events: many studies have shown that statins reduce CVD-related morbidity and mortality in patients with or without prior CVD [1–3]. However, the questions about their harmful effects and beneficiaries, particularly in Asian populations, have remained unresolved.” In general the harmful effects and beneficiaries of statins is well studied. In Asian populations you can also find quite much evidence. This should be clarified here. 

Response: Thank you for your comments. I acknowledge that there have been many observational studies that reported the increased risk of diabetes in Asian statin users and a clinical trial for the secondary prevention of coronary artery disease in Japanese patients (https://doi.org/10.1161/CIRCULATIONAHA.117.032615). I have removed the words “particularly in Asian populations” in that sentence and added the cohort studies in primary prevention as references (#23 and #24) in the 2nd paragraph of the Discussion. 

In conclusion section (in main manuscript and in abstract) should include the specific population of the study (Asians). 

Response: I have included the specific population of the study in the Conclusion of the Abstract and in the Conclusion of the Text. 

Specific comments:

Page 6 and other pages and tables: “Never use statins” group should be named as “non-users” in all section since the group includes patients not using statins over 90 days/year, but not never. Non-users term is used in result section, and I assume it means the same group. If this is not the case please clarify.

Response: To clarify that, I have used the term “nonuse” rather than “never use” in the manuscript.

Page 7. Body mass index group 10.0-18.5. Are there patients having BMI 10.0-16? I would group these extremely anorectic individuals in different group (or exclude). At least some comment on this would be needed.

Response: In raw data of the NHID, there existed an extreme value of variables (e.g., BMI <10.0) although it was very rare.The extreme values were deleted before analyses as those would be resulted from recording mistakes. The BMI values of 10.0-16.0 were also rare, and it was not sure whether the values were true or recording errors. In this study, BMI values were not included in the calculation of CVD risk scores and were categorized before the analysis. I believe that the exclusion or further categorization of the low BMI would change the results very little. However, the Limitations paragraph has been revised to include the possibility of inadequate matching (adjustment). 

Page 7: dinking means alcohol consumption? 

Response: To avoid confusion, I have replaced the term with “alcohol consumption” in the manuscript.

In discussion section, please specify if you talk about CVD mortality or all cause mortality. 

Response: I have revised the manuscript to specify CVD risk and all-cause mortality.

Page 15. “Recent meta-analyses of clinical trials found no or only minimal associations between statin use and diabetes risk [2,20].” Recent is not 10 years ago.

Response: I have specified the years of the publications as “In 2015 and 2016 meta-analyses...” in that sentence. 

Page 16. “However, they disapproved of the arguable role of risk calculators in determining optimal blood pressure for antihypertensive treatment.” I would recommend removing this sentence, it is not clear that you are referring other studies of yours here. If you like to keep this ad reference and clarify, maybe change the place for this sentence before you state that “Overall, these findings strongly support the validity of CVD risk calculators indetermining potential beneficiaries of statin use.” And then just use, “In this study, these findings…” And dont use "they" if you mean you. Take credits of your own work!

Response: The CVD risk calculators are currently recommended to determine candidates for statin therapy or candidates for intensive BP-lowering. I have revised the sentence to “The present findings strongly support the validity of CVD risk calculators in determining potential beneficiaries of statin use. However, this study does not ensure the role of risk calculators in determining optimal blood pressure for antihypertensive treatment.”

---

## [Editor Report · Decision Letter 1]

5 Jan 2021

Statin use and outcome risks according to predicted CVD risk in Korea: A retrospective cohort study

PONE-D-20-35305R1

Dear Dr. Jung,

We’re pleased to inform you that your manuscript has been judged scientifically suitable for publication and will be formally accepted for publication once it meets all outstanding technical requirements.

Kind regards,

Katriina Aalto-Setala, Professor

Academic Editor

PLOS ONE
---

## [Editor Report · Acceptance letter]

7 Jan 2021

PONE-D-20-35305R1 

Statin use and outcome risks according to predicted CVD risk in Korea: A retrospective cohort study 

Dear Dr. Jung:

I'm pleased to inform you that your manuscript has been deemed suitable for publication in PLOS ONE. Congratulations! Your manuscript is now with our production department. 

Kind regards, 

on behalf of

Dr Katriina Aalto-Setala 

Academic Editor

PLOS ONE